# Burden of undiagnosed hypertension and its associated factors: A challenge for primary health care in urban Colombia

**Jorge Emilio Salazar Flórez**[1]*, **Ángela Patricia Echeverri Rendón**[2], **Luz Stella Giraldo Cardona**[1]

**1** Research Center, Grupo de Estudio de Enfermedades Infecciosas y Crónicas (GEINCRO), San Martin University Foundation, Sabaneta, Colombia, **2** Grupo de Estudio de Enfermedades Infecciosas y Crónicas (GEINCRO), San Martin University Foundation, Sabaneta, Colombia

* jorge.salazarf@sanmartin.edu.co

## Abstract

### Background

Arterial hypertension is one of the most prevalent chronic, non-communicable diseases and the leading preventable risk factor for cardiovascular disease (CVD) and all-cause mortality worldwide. Although its primary causes and consequences are preventable, it often remains undiagnosed. Consequently, this study aims to determine the prevalence and factors associated with normotensive, diagnosed, and undiagnosed hypertension in adults.

### Methods

A cross-sectional, population-based study was conducted in Sabaneta, Colombia, between 2021 and 2022, with 286 adults aged 18 and older. Stratified and systematic random sampling methods were employed. The World Health Organization STEP survey and the Perez Rojas test were utilized to assess behavioral risk factors and sedentary lifestyles. Body mass index, waist circumference, and arterial tension were measured using standardized instruments. The prevalence of hypertension was then estimated. Risk factors influencing normotensive, diagnosed, and undiagnosed hypertension were analyzed using multinomial regression. The outcome variable comprised three categories: normotensive (reference category), diagnosed hypertension, and undiagnosed hypertension. The multinomial regression coefficients were exponentiated and are presented as relative risk ratios (RRR) with 95% confidence intervals (CI). The model was adjusted for sex and sample weight per neighborhood.

### Results

The study revealed a hypertension prevalence of 38.5% and an undiagnosed hypertension rate of 50.9%. Those with undiagnosed hypertension were predominantly adults over 60 years (RRR = 0.68; 95% CI: 0.53–0.86), individuals with an elementary school education (RRR = 1.75; 95% CI: 1.27–2.42), those physically active (RRR = 1.52; 95% CI: 1.22–1.89), without prior diagnoses of chronic comorbidities (RRR = 1.42; 95% CI: 1.12–1.82), and with

**Funding:** This study was funded by the San Martin University Foundation (PYI-2020-025).

**Competing interests:** The authors have declared that no competing interests exist.

obesity (RRR = 2.25; 95% CI: 1.63–3.11) or overweight conditions (RRR = 1.70; 95% CI: 1.334–2.15).

## Conclusions

Undiagnosed hypertension was significant among populations without risk conditions. There is an urgent need for community-based early detection and education strategies to mitigate this issue.

## Introduction

Hypertension (HTN), a dominant chronic condition [1], serves as the primary modifiable risk factor for global cardiovascular disease and all-cause mortality [2]. Defined by a systolic BP ($\geq$140 mmHg) or diastolic BP ($\geq$90 mmHg), HTN was linked to 10.4 million deaths in 2017, affecting 20% of adults worldwide [1–3]. This disease is shaped by genetics, age, substance use, lifestyle, healthcare accessibility, and dietary habits [1, 4–7].

Financially, HTN and its related ailments burden health systems. In 2001, high BP costs approximated 10% of the global health expenditures [8]. In Colombia, basic HTN care costs 184,631 pesos per patient monthly, but complications can substantially increase this figure [9]. In low- and middle-income countries (LMICs), HTN prevalence surpasses that of affluent nations, yet diagnosis and treatment lag [1, 5]. Remarkably, 75% of those with HTN are in LMICs [1, 6]. Recent studies show BP decreasing in high-income countries over four decades, whereas it rises in LMICs, notably in Latin America [7, 10].

In 2019, CVD was the primary cause of death in the Americas, responsible for one-third of regional fatalities. The same year recorded hypertension in 34.4% of the populace, with effective control in merely 40.9% of women and 32.2% of men [11–13]. A significant number remain undiagnosed due to asymptomatic nature of hypertension, reinforcing the "hypertension care gap" [4]. The challenge lies in early diagnosis, treatment adherence, and control to diminish cardiovascular mortality [11, 12, 14]. Disturbingly, 46% of hypertensive individuals are undiagnosed [15], and in a sample of adults aged 15–54, 66.8% were similarly undiagnosed [16]. Several studies pinpoint male gender, tobacco use, physical activity, and being overweight as key risk factors in undiagnosed hypertension, especially evident in Asian and African nations with rates from 50.4% to 59.9% [14, 15, 17–19].

Research into care cascade gaps and undiagnosed hypertension remains scant. Data from the 2010 CARMELA study indicated undiagnosed hypertension rates from 24.0% to 47.0% across seven Latin American cities [20], while the U.S. had a rate below 7.0% in 2018 [21]. A 2021 study in Peru unveiled a 67.0% prevalence [22]. However, there is a distinct lack of data for Colombia and its neighbors, leading to the focus of this study, which aims to identify risk factors of undiagnosed hypertension in Colombian adults and guide policymakers in tailored strategy development.

## Materials and methods

This research is a population-based cross-sectional study. The reporting of results adhered to the STROBE guidelines for observational studies (S1 Checklist). Stratified sampling was employed based on three criteria: 1) area of residence: rural or urban; 2) age group: young adult (<41 years), middle-aged adult (41–60 years), and older adult (>60 years); and 3) sex:

male or female. To target the units of analysis (adults aged over 18 years), a systematic random selection was enforced in each neighborhood or sidewalk, opting for every second household. This study spanned from February 01, 2022, to December 20, 2022, in Sabaneta, Colombia.

To determine the sample size, we referenced the 2019 population projection from the National Administrative Department of Statistics (DANE, as per its Spanish acronym). In 2019, the population aged over 18 in the municipality totaled 69,045 individuals [23]. A formula was utilized to project a prevalence, setting a 95.0% confidence interval, an anticipated hypertension prevalence of 28.0%, and a maximum permissible error of 5.0%. This resulted in a targeted sample of 306 individuals. The computation was executed using the free online software, Open Epi 3.01. The study ultimately encompassed 286 adults, translating to a response rate of 93.5%.

## Instruments

Interviews were conducted face-to-face by medical students who were both qualified and experienced in data collection. Information was gathered on demographics, anthropometric indicators, sedentary behaviors, behavioral risk factors, arterial tension, and heart rate.

Vital signs of all participants were checked three times. Heart rate was determined using an oximeter (model MD300C29) after a 15-minute rest. Arterial tension was assessed using a pre-calibrated manometer (model CE 0297). The procedure for measuring arterial pressure adhered to the guidelines of the Sixth Joint National Committee (JNC VI) [24] which is in line with the recommendations of the American Heart Association [25], the American Society of Hypertension, and the Pan American Health Organization (PAHO) [26]. After a 15-minute rest period, enumerators took three readings each of systolic and diastolic blood pressures at five-minute intervals. An individual was classified as hypertensive if they recorded a systolic blood pressure (SBP) of ≥140 mmHg and/or a diastolic blood pressure (DBP) of ≥90 mmHg, as per the JNC VI guidelines [24]. Undiagnosed hypertension was characterized by having an SBP of ≥140 mmHg or a DBP of ≥90 mmHg, without any prior hypertension diagnosis from a healthcare professional or without being on antihypertensive medications. The study's primary variable of interest was segmented into three categories: normotensive, diagnosed hypertension, and undiagnosed hypertension.

Anthropometric measurements were taken using various instruments. A calibrated balance model 142KL was used for weight determinations, and for height, a tallimeter with a movable foot was employed. Abdominal and pelvic circumferences were measured with a handheld tape measure.

The World Health Organization (WHO) Chronic Diseases STEPwise approach to Surveillance (STEPS) survey was incorporated, designed to assess behavioral, anthropometric, biological, and demographic attributes [27]. Pertinent to our study, we extracted demographic details such as gender, age, marital status, and education level. Behavioral risk factors considered included smoking, alcohol consumption, and fruit and vegetable intake. Cardiometabolic risk factors taken into account were high body mass index, diabetes, and elevated cholesterol.

Additionally, the Pérez-Rojas-García test for classifying sedentary lifestyles was administered [28]. Participants were stratified as either severely sedentary, moderately sedentary, active, or highly active based on the test outcomes. The process began with recording the heart rate. Participants were then subjected to three bouts of exercise at varying intensities, each bout involving stepping up and down on a 25 cm step for three minutes with a one-minute rest interval between each. The set intensities were 68, 104, and 144 beats per minute for the respective bouts. If, after the initial bout, the heart rate remained below 120 bpm, the subsequent bout was initiated. An early termination of the test occurred if the heart rate exceeded

120 bpm following the first bout, categorizing the individual as "severely sedentary." Those who completed the second bout were classified as "moderately sedentary." This instrument had previously been validated on the Colombian population aged between 18 and 60 years [29].

## Statistical analysis

Data analysis was conducted using R software (v 4.2.2, www.r-project.org/). The demographic attributes, lifestyle habits, anthropometric indices, and comorbidities were presented through absolute frequencies and percentages for qualitative variables, while means and standard deviations described the quantitative variables. Data was segmented based on three categories: normotensive, diagnosed hypertension, and undiagnosed hypertension. Differences across these categories in terms of general characteristics, lifestyle, and comorbidities were assessed via the chi-square test of independence. All statistical tests were two-tailed with a significance level set at α = 0.05.

For understanding the relationship between dependent and independent variables, a multivariable multinomial regression model was employed, adjusting for potential confounders. Every variable was introduced into the multinomial logistic regression model in one step, and the least significant ones were sequentially removed until the most concise model was achieved. The dependent variable had three classifications: normotensive, diagnosed hypertension, and undiagnosed hypertension, with the normotensive group serving as the reference for comparison in the regression analysis. Coefficients from the multinomial regression were exponentiated and presented as relative risk ratios (RRR) with their corresponding 95% confidence intervals (CI). Here, the RRR represents the likelihood of an outcome in the exposed group relative to its occurrence in the unexposed group. In the context of this study, a p-value less than 0.05 indicated statistical significance. Independent variables encompassed aspects like lifestyle choices, substance use, dietary habits, anthropometric indicators, sedentary behavior, and existing comorbidities. Adjustments in this model were made for gender and a weighting variable representing the distribution of participation across neighborhoods. The choice of variables incorporated in the multinomial model was driven by their theoretical relevance, as depicted in Fig 1. Existing research posits that inconsistencies in hypertension care are unevenly spread across different demographic sections. This implies that younger

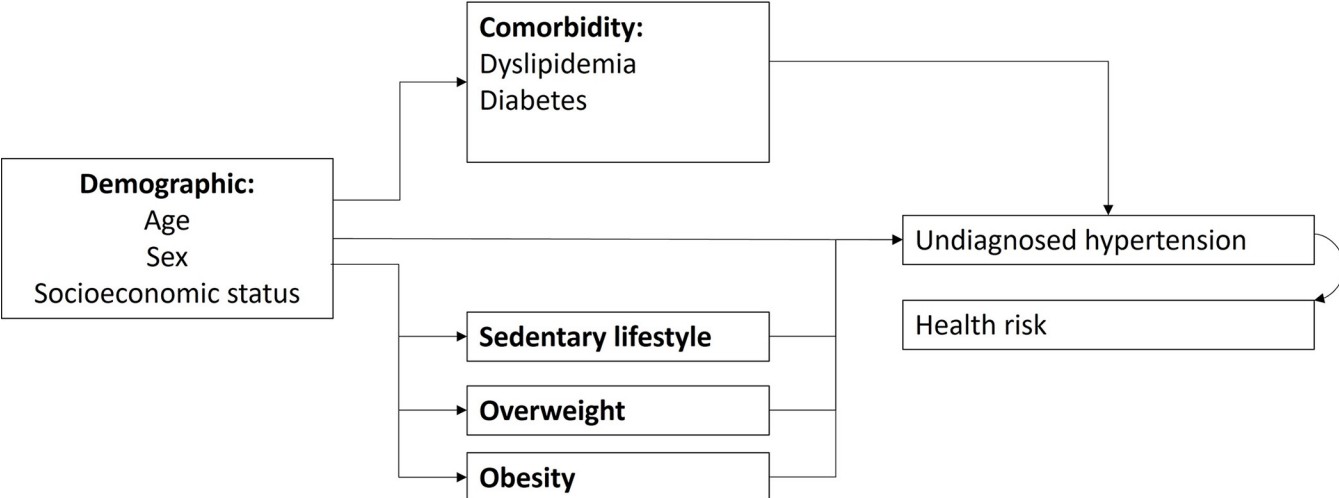

**Fig 1. A theoretical model for the relationship between undiagnosed hypertension and risk factors in adults over 18 years.** Source: created by the authors.

individuals are notably more susceptible to inadequate hypertension management [15, 22, 30]. Additionally, groups like women, economically disadvantaged individuals, those with minimal educational backgrounds, those adopting risky lifestyles (e.g., unhealthy eating habits, limited physical activity, alcohol consumption), and rural dwellers are more inclined to either be oblivious of their hypertension, remain untreated, or have uncontrolled hypertension [18, 30]. Such observations align with the hypothesis posited by Dhungana et al. [4], emphasizing that any lapse in the care continuum augments health risks.

This study adhered to all international research protocols concerning human subjects and the Colombian standard 8430 established in 1993 [31]. Participation was entirely voluntary, with participants providing informed consent. The research received prior approval from the Research Center and Ethical Review Board of the Sabaneta campus of San Martin University, under the reference CI-FUSM-2022-001. Before initiating the data collection process, written consent was obtained from the participants. Furthermore, the tools used for data collection ensured full confidentiality, safeguarding the privacy of the participants. The authors were never privy to any information that could personally identify the participants, both during the study and post data collection.

## Results

Of the participants, 38.5% had hypertension (95.0% CI: 32.7–44.3). Out of the 110 hypertensive participants, 50.9% were undiagnosed (95.0% CI: 41.1–60.7) as depicted in Fig 2.

The demographics of the participants are summarized in Table 1. The majority of participants were female (n = 163; 57.0%) and below the age of 60 (n = 183; 63.9%). Participants' ages ranged from 19 to 91 years, with a median age of 51.2 years, an interquartile range of 34.3 years, a mean age of 50.2 years, and a standard deviation of 18.8 years. Over 80% of the participants accessed health care either through the contributory scheme or as beneficiaries, which is how the salaried population in Colombia typically accesses health care services. Regarding

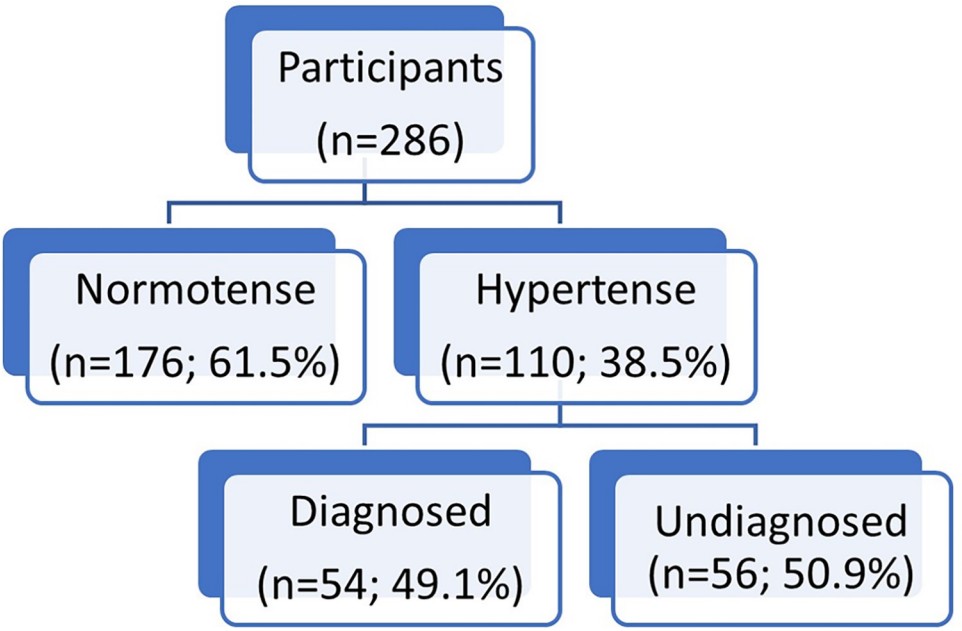

**Fig 2. Prevalence of normotensive, diagnosed and undiagnosed hypertension in adults over 18 years.** Sabaneta, 2022. *Source*: *created by the authors*.

**Table 1. Demographic characteristics in adults over 18 years by normotensive, diagnosed and undiagnosed hypertension.** Sabaneta, 2022.

| Factor | Normotensive | | Diagnosed hypertension | | Undiagnosed hypertension | | Total | | p value[a] |
|---|---|---|---|---|---|---|---|---|---|
| | n = 176 | % | n = 54 | % | n = 56 | % | n = 286 | % | |
| **Sex** | | | | | | | | | |
| Female | 101 | 62.0 | 32 | 19.6 | 30 | 18.4 | 163 | 57.0 | 0.822 |
| Male | 75 | 61.0 | 22 | 17.9 | 26 | 21.1 | 123 | 43.0 | |
| **Age (Years)** | | | | | | | | | |
| < 60 | 120 | 65.6 | 29 | 15.8 | 34 | 18.6 | 183 | 64.0 | 0.130 |
| > = 60 | 56 | 54.4 | 25 | 24.3 | 22 | 21.4 | 103 | 36.0 | |
| **Social protection in health** | | | | | | | | | |
| Beneficiary / Contributor | 135 | 62.2 | 39 | 18.0 | 43 | 19.8 | 217 | 75.9 | 0.409 |
| Linked / Other | 24 | 60.0 | 11 | 27.5 | 5 | 12.5 | 40 | 14.0 | |
| **Education level** | | | | | | | | | |
| Elementary school | 65 | 53.7 | 30 | 24.8 | 26 | 21.5 | 121 | 42.3 | 0.243 |
| High school | 57 | 66.3 | 11 | 12.8 | 18 | 20.9 | 86 | 30.1 | |
| Technical or technological | 25 | 71.4 | 6 | 17.1 | 4 | 11.4 | 35 | 12.2 | |
| University / postgrad | 29 | 65.9 | 7 | 15.9 | 8 | 18.2 | 44 | 15.4 | |
| **Marital status** | | | | | | | | | |
| Single | 76 | 67.9 | 17 | 15.2 | 19 | 17.0 | 112 | 39.2 | 0.215 |
| Married / partnership | 77 | 59.2 | 29 | 22.3 | 24 | 18.5 | 130 | 45.5 | |
| Separated / divorced / widowed | 23 | 52.3 | 8 | 18.2 | 13 | 29.5 | 44 | 15.4 | |
| **Socioeconomic status** | | | | | | | | | |
| Low | 99 | 62.3 | 30 | 18.9 | 30 | 18.9 | 159 | 55.6 | 0.555 |
| Middle | 67 | 58.3 | 23 | 20.0 | 25 | 21.7 | 115 | 40.2 | |
| **Sample** | **176** | | **54** | | **56** | | **286** | | |

a: chi-square test of association

education, 42.3% (n = 121) had finished elementary school, while 30.1% had completed high school. In terms of marital status, the largest segment was either married or in a partnership (n = 130; 45.5%). Socioeconomically, the majority (n = 159; 58.0%) were categorized as low income. When comparing demographic characteristics among normotensive, diagnosed, and undiagnosed hypertension groups, no significant differences were observed (p-value > 0.05).

Approximately 51.4% (n = 147) of the participants were aware of the health benefits plan. A significant 60.0% (n = 172) reported alcohol consumption. Only 37.4% consumed the recommended amount of fruit, while 50.3% met the recommended vegetable intake. When looking at those diagnosed with hypertension, 49.4% had dyslipidemia, and 42.1% led a sedentary lifestyle. Additionally, 42.0% of the total respondents were overweight. Notably, within the hypertensive group, there was a higher prevalence of obesity at 56.0% and abdominal obesity at 50.9% (as detailed in Table 2). Based on the data in Table 2, significant statistical differences were identified in the prevalence of hypertension in relation to diabetes (p = 0.003), dyslipidemia (p = 0.001), heart attack history (p = 0.036), sedentary habits (p = 0.001), BMI (p = 0.009), and abdominal obesity (p = 0.000).

The results from adjusted multinomial logistic regression models for predicting both diagnosed and undiagnosed hypertension, compared with normotensive, are presented in Table 3. Individuals under 60 were 55.0% less likely to be diagnosed with HTN than older adults (RRR = 0.45 [95% CI: 0.35–0.57]). This age group was also less likely to have undiagnosed hypertension (RRR = 0.68 [95% CI: 0.53–0.86]). Possessing prior comorbidities and being

**Table 2. Clinical history and lifestyle in adults older than 18 years by normotensive, diagnosed and undiagnosed hypertension.** Sabaneta, 2022.

| Factor | Normotensive | | Diagnosed hypertension | | Undiagnosed hypertension | | Total | | p value[a] |
|---|---|---|---|---|---|---|---|---|---|
| | n = 176 | % | n = 54 | % | n = 56 | % | n = 286 | % | |
| **Knowledge of health programs** | | | | | | | | | |
| Health benefits plan | 99 | 67.3 | 23 | 15.6 | 25 | 17.0 | 147 | 51.4 | 0.134 |
| Cardiovascular program | 44 | 64.7 | 15 | 22.1 | 9 | 13.2 | 68 | 23.8 | 0.431 |
| **Use of substance** | | | | | | | | | |
| Psychoactive | 23 | 69.7 | 4 | 12.1 | 6 | 18.2 | 33 | 11.5 | 0.735 |
| Tobacco | 26 | 70.3 | 7 | 18.9 | 4 | 10.8 | 37 | 12.9 | 0.334 |
| Alcohol | 114 | 66.3 | 27 | 15.7 | 31 | 18.0 | 172 | 60.1 | 0.109 |
| **Food** | | | | | | | | | |
| Vegetable oil | 162 | 61.8 | 48 | 18.3 | 52 | 19.8 | 262 | 91.6 | 0.653 |
| At least 7 fruits a week | 63 | 58.9 | 21 | 19.6 | 23 | 21.5 | 107 | 37.4 | 0.571 |
| At least 7 vegetables per week | 92 | 63.9 | 28 | 19.4 | 24 | 16.7 | 144 | 50.3 | 0.432 |
| **Comorbidity** | | | | | | | | | |
| Diabetes | 16 | 44.4 | 16 | 44.4 | 4 | 11.1 | 36 | 12.6 | 0.003 |
| Dyslipidemia | 41 | 50.6 | 29 | 35.8 | 11 | 13.6 | 81 | 28.3 | 0.001 |
| Heart attack | 13 | 59.1 | 8 | 36.4 | 1 | 4.5 | 22 | 7.7 | 0.036 |
| **Sedentary** | | | | | | | | | |
| Active | 61 | 70.9 | 4 | 4.7 | 21 | 24.4 | 86 | 30.1 | 0.001 |
| Sedentary | 92 | 57.9 | 36 | 22.6 | 31 | 19.5 | 159 | 55.6 | |
| **BMI** | | | | | | | | | |
| Normal | 83 | 71.6 | 16 | 13.8 | 17 | 14.7 | 116 | 40.6 | 0.009 |
| Overweight | 71 | 59.2 | 23 | 19.2 | 26 | 21.7 | 120 | 42.0 | |
| Obesity | 22 | 44.0 | 15 | 30.0 | 13 | 26.0 | 50 | 17.5 | |
| **Abdominal obesity** | 52 | 49.1 | 33 | 31.1 | 21 | 19.8 | 106 | 37.1 | 0.000 |

a: chi-square test of association

BMI: Body Mass Index

**Table 3. A multinomial regression model of associated factors of diagnosed and undiagnosed hypertension in adults over 18 years.** Sabaneta, 2022.

| Factor | Diagnosed hypertension vs. Normotensive | | | Undiagnosed hypertension vs. Normotensive | | |
|---|---|---|---|---|---|---|
| | RRR | 95% CI—RRR | p value | RRR | 95% CI—RRR | p value |
| < 60 years (ref: >60) | 0.45 | 0.35–0.57 | 0.000 | 0.68 | 0.53–0.86 | 0.001 |
| **Education level (ref: University / postgrad)** | | | | | | |
| Elementary school | 1.83 | 1.26–2.67 | 0.002 | 1.75 | 1.27–2.42 | 0.001 |
| High school | 2.03 | 1.34–3.00 | 0.000 | 1.33 | 0.95–1.86 | 0.094 |
| Technical or technological | 1.25 | 0.79–1.98 | 0.334 | 0.86 | 0.57–1.30 | 0.484 |
| **Previous diagnosis of dyslipidemia or diabetes (ref: Yes)** | 0.45 | 0.36–0.56 | 0.000 | 1.42 | 1.12–1.82 | 0.004 |
| **Active (Perez Rojas Test) (ref: inactive)** | 0.35 | 0.26–0.47 | 0.000 | 1.52 | 1.22–1.89 | 0.000 |
| **BMI (ref: normal)** | | | | | | |
| Obesity | 2.49 | 1.83–3.40 | 0.000 | 2.25 | 1.63–3.11 | 0.000 |
| Overweight | 1.12 | 0.86–1.45 | 0.412 | 1.70 | 1.34–2.15 | 0.000 |

Adjusted by sex—AIC for adjusted model: 4681,305—AIC for null model: 6462,292—LR chi2(20) = 528.90; prob>chi2 = 0.000; pseudo R2 = 0.1024—BMI: Body Mass Index -

physically active decreased the probability of being diagnosed with HTN by 55% and 65%, respectively. Those with elementary and high school education had significantly higher rates of both diagnosed and undiagnosed hypertension than those with university/postgraduate degrees. BMI plays a crucial role in the diagnosis and presence of undiagnosed hypertension. Specifically, having obesity increased the risk of undiagnosed hypertension by 1.25 times (RRR = 2.25 [95% CI: 1.63–3.11]), while being overweight amplified the risk by 70% (RRR = 1.70 [95% CI: 1.34–2.15]). Lastly, having a prior comorbidity and being physically active raised the chances of having undiagnosed hypertension by 42% and 52%, respectively.

## Discussion

The study aimed to estimate the prevalence of undiagnosed hypertension and its associated factors in individuals aged 18 and above in a Colombian municipality. The prevalence rates for hypertension and undiagnosed hypertension stood at 38.5% and 50.9%, respectively. In the present study, individuals who had completed either elementary or high school, had no chronic diseases, were physically active, and were either obesity or overweight were more inclined to be unaware of their hypertensive status.

The prevalence observed in this study (38.5%) surpassed both the national and global averages of 24.0% and 30.0% respectively [3, 11, 32], but was consistent with rates in Latin America. Research from 2019, encompassing 33,276 participants from six Latin American nations, reported an HTN prevalence of 44.0%. The lowest rates were found in Peru (17.7%) and the highest in Brazil (52.5%); further, a mere 58.9% were cognizant of their HTN diagnosis [33].

Regarding the prevalence of undiagnosed hypertension (50.9%) among the hypertensive populace, discrepancies emerge between studies that depend on clinical registries or surveillance systems. This is evident when comparing data from the U.S. (7.0% undiagnosed HTN) [21] and China (28.8%) [34]. However, these figures align closely with findings from representative, population-based studies typically conducted outside the Americas, which reported between 50.4% to 69.0% undiagnosed HTN [16–19]. This was on par with results from Malaysia (51.6%) [17] and Bangladesh (50.1%) [19]; however, there was a notable discrepancy in Africa (69.8%) [18], and nearly double the prevalence in Nepal (34.1%) [4]. Our results surpass recent data on undiagnosed hypertension in Latin America (which ranged from 24% to 47%) [20], yet remain below Peru's reported figure (67.0%) [22]. It becomes evident that roughly half of individuals with hypertension remain oblivious to their condition, a phenomenon termed the "rule of halves", which holds true across various populations.

Variability in the type of sample or source data might account for the differences observed in the prevalence of both hypertension and its undiagnosed counterpart. Many nationwide surveys in Latin America rely on self-reports or lack serial blood pressure (BP) measurements, potentially leading to underestimation of prevalence rates [5, 33]. Truly population-based estimates are scant [18, 19]. Often, such estimates lean on secondary sources like clinical registries, epidemiological surveillance data, or national surveys (e.g., demographic and health surveys) [14, 15, 16, 17, 22, 35], each tailored for unique objectives. Such sources might not aptly represent the at-risk population.

There are other conceivable factors contributing to the elevated prevalence of undiagnosed HTN. One such factor is the general unawareness surrounding routine blood pressure screening. The frequency of BP measurements directly correlates with awareness of hypertension, a crucial element for diagnosis [4, 19]. Evidently, and as this study demonstrates, the prevalence of undiagnosed hypertension remains stubbornly high over recent decades [12, 20, 22]. The stagnant state of HTN awareness and screening over the years is alarming. It underscores the inadequacy of current methods used to detect HTN, even though BP measurements are

straightforward and cost-effective. Addressing arterial hypertension is an ongoing challenge in this region [12], with Primary Health Care (PHC) being pivotal in early detection. Public health initiatives should prioritize the prevention, detection, treatment, and comprehensive cardiovascular risk management.

While the broader population should be the primary target for HTN detection and management within primary health care, our findings indicate that specific subgroups -notably those without chronic diseases, who are physically active, have only an elementary education, or had obesity or overweight—are at a heightened risk for undiagnosed HTN. This calls for an intensified focus on these demographics. Earlier studies have identified similar risk factors [18, 22, 34]. Such trends could be attributed to the likelihood of individuals with chronic ailments or comorbidities seeking medical care more frequently, thus increasing the odds of an HTN diagnosis [13, 18]. Conversely, highly active individuals might perceive themselves as less susceptible to chronic illnesses, thereby paying less attention to their health [14]. Some research [14, 18] has even established a positive link between intense physical activity and undiagnosed HTN.

In alignment with the findings of this study and various other studies [14, 19, 22, 34], factors linked to healthcare needs and inversely associated with undiagnosed HTN include conditions like dyslipidemia and diabetes. Individuals without chronic conditions tend to access health services less frequently, thereby reducing the chances of undergoing a blood pressure assessment [1, 6]. On the other hand, while those who are overweight might recognize their health issues [1, 6, 18], they might view health facilities as judgmental environments, leading them to steer clear of medical counsel. A population-centric study highlighted that individuals who are overweight or who suffer from obesity are doubly prone to remain undiagnosed [36].

Previous studies have been ambivalent about the influence of education on the diagnosis of hypertension [36]. Yet, our research identified a heightened risk of undiagnosed hypertension among individuals with only elementary education (when juxtaposed against those with professional qualifications). Certain researchers have pointed out that less educated individuals tend to have a greater prevalence of undiagnosed hypertension [17]. This discrepancy can potentially be attributed to disparities in health education access, risk perception, and the ease with which higher-educated individuals can access health services.

In conclusion, while hypertension is widespread among the elderly, its undiagnosed variant tends to surge with advancing age. Echoing prior studies, our data reaffirmed that older individuals are at an amplified risk of unrecognized hypertension [14, 18, 36]. Nonetheless, some researchers have indicated that younger adults frequently remain oblivious of their hypertensive condition [18]. Past studies have posited that those aged 50 and above exhibit a diminished prevalence of undiagnosed hypertension [4, 15, 22]. Contrarily, our findings suggest that advancing age diminishes the chances of undiagnosed hypertension. This sentiment has found support in other research, which documented a risk decrement of roughly 35% [18]. Furthermore, investigations employing multinomial models (with normotensive individuals as the reference category) have deduced that adults younger than 30 are less prone to undiagnosed hypertension [36].

Existing research indicates that recent interactions with health services, possession of health insurance, or undergoing health screenings correlate with a reduction in undiagnosed HTN cases [4, 15, 18, 22, 34]. Engaging with health services can potentially optimize outcomes throughout the hypertension care process [4], encompassing the stages of hypertension screening, awareness, treatment, and management. Nonetheless, our current study does not provide data to elucidate this association.

There is mounting evidence that contemporary prevention and control tactics are coming up short. A more strategic, holistic, and long-lasting solution is urgently required to diminish

the toll of CVD and bolster the resilience of healthcare infrastructures. PHC emerges as a pivotal force in streamlining the hypertension care continuum [12, 37]. In LMICs, including countries like Colombia, there's a noticeable deficit in preventive and screening cultures, leading people to seek medical attention predominantly when symptoms manifest. This could explain why these nations report an alarming 75.0% prevalence rate of hypertension [1, 6]. Health models rooted in the PHC approach could serve as an effective conduit to early hypertension detection services. A notable initiative in the Americas in this context is the introduction of the HEARTS Project [12, 37].

Initiatives such as the HEARTS program underscore a paradigm shift. In this renewed model, hypertension management and CVD prevention aren't exclusively tethered to secondary or tertiary care tiers but are integrated into the primary care framework [12, 37]. The program also champions an educational model for the PHC team, ensuring they possess a thorough understanding of their communities, patients, and surrounding circumstances. Additionally, it introduces innovative engagement methods, like telemonitoring [12, 25, 38]. Yet, this model is not without its challenges, particularly in reaching out to those who do not actively seek medical services. Consequently, it is imperative to extend beyond traditional medical facilities, fostering strong community ties and embracing localized health narratives by incorporating community liaisons and external strategies. A uniform methodology for participant selection, blood pressure assessment, data quality assurance, and analysis is equally crucial. In the final analysis, there's an urgent need for research endeavors to devise and deploy impactful, fair, and enduring evidence-informed interventions and global health policies, aiming squarely at mitigating the overarching impact of hypertension [6].

The study presented several limitations. Firstly, due to the cross-sectional design, it wasn't possible to infer causality. Secondly, relying on self-reported data, such as diagnosed hypertension, might introduce recall bias. Thirdly, family history was not factored in as a covariate. Fourthly, blood pressure in the study was measured three times during physical examinations, deviating from the Hypertension Clinical Practice Guidelines that recommend an average of ≥2 readings obtained on ≥2 separate occasions [25]. Fifthly, although the desired sample size was not achieved, it is noteworthy that the attrition rate stood at a modest 6.5% (n = 20). Additionally, considering the actual hypertension prevalence observed in this study (38.5%), the precision amounted to 5.6. Thus, the concluding sample of 286 participants did not markedly compromise the estimate's accuracy, an essential metric in prevalence research. Regardless of these constraints, our study carved a niche for itself. It utilized a population-based sample that mirrors the adult demographic across the municipality, meticulously analyzed the risk factors tied to undiagnosed hypertension, and furnished insights to guide forthcoming hypertension prevention initiatives.

To wrap up, the prevalence rates of hypertension and its undiagnosed variant remain alarmingly high, posing considerable challenges not just to Colombia but also to the broader Americas. There is an urgent call for national and regional public health campaigns to augment hypertension detection, raise awareness, and bolster its treatment and management in Colombia and across Latin America. By directing mass screening efforts towards the demographics most affected, enhancing access to quality care within primary public health establishments, and rolling out community-based interventions, we can effectively tackle the obstacles impeding early hypertension screening.

## Supporting information

**S1 Checklist. STROBE statement—Checklist.**
(DOCX)

## Author Contributions

**Conceptualization:** Jorge Emilio Salazar Flórez, Ángela Patricia Echeverri Rendón.

**Data curation:** Jorge Emilio Salazar Flórez, Ángela Patricia Echeverri Rendón.

**Formal analysis:** Jorge Emilio Salazar Flórez, Ángela Patricia Echeverri Rendón, Luz Stella Giraldo Cardona.

**Investigation:** Jorge Emilio Salazar Flórez.

**Methodology:** Jorge Emilio Salazar Flórez, Luz Stella Giraldo Cardona.

**Project administration:** Jorge Emilio Salazar Flórez, Ángela Patricia Echeverri Rendón.

**Writing – original draft:** Jorge Emilio Salazar Flórez, Ángela Patricia Echeverri Rendón, Luz Stella Giraldo Cardona.

**Writing – review & editing:** Jorge Emilio Salazar Flórez, Ángela Patricia Echeverri Rendón, Luz Stella Giraldo Cardona.

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
