## [Decision Letter · Decision Letter 0]

5 Jul 2023

PONE-D-23-08509BURDEN OF UNDIAGNOSTIC HYPERTENSION ANN ASSOCIATED FACTORS: A CHALLENGE FOR PRIMARY HEALTH CAREPLOS ONE

Dear Dr. Salazar Flórez,

Thank you for submitting your manuscript to PLOS ONE. After careful consideration, we feel that it has merit but does not fully meet PLOS ONE’s publication criteria as it currently stands. Therefore, we invite you to submit a revised version of the manuscript that addresses the points raised during the review process. The manuscript has been reassessed by one reviewer. The comments are appended below. The reviewer observe major concerns about the manuscript, and in particular they feel that important methodological issues exist that affect the technical soundness of your study, and the conclusions of the paper. We request authors to conduct a detailed review of the comments. 

We look forward to receiving your revised manuscript.

Kind regards,

Bruno Pereira Nunes, Ph.D.

Academic Editor

PLOS ONE

Journal Requirements:

2. Please provide additional details regarding participant consent. In the Methods section, please ensure that you have specified (1) whether consent was informed and (2) what type you obtained (for instance, written or verbal). If your study included minors, state whether you obtained consent from parents or guardians. If the need for consent was waived by the ethics committee, please include this information.

This study was funded by the San Martin University Foundation (PY-2020-025).

Reviewers' comments:

Reviewer's Responses to Questions

**Comments to the Author**

1. Is the manuscript technically sound, and do the data support the conclusions?

Reviewer #1: No

2. Has the statistical analysis been performed appropriately and rigorously? 

Reviewer #1: No

3. Have the authors made all data underlying the findings in their manuscript fully available?

Reviewer #1: Yes

4. Is the manuscript presented in an intelligible fashion and written in standard English?

Reviewer #1: No

5. Review Comments to the Author

Reviewer #1: Although the manuscript focused on a topic of public health importance, it needs major revision as follows:

1. The English language needs professional editing throughout the manuscript.

2. The title does not make sense. The word “undiagnostic” in the title should be replaced by "undiagnosed". The title should include the geographical area of the study population. Suggested title may be "Burden of undiagnosed hypertension and its associated factors: A challenge for primary health care in urban Colombia"

3. The words “undiagnosed” and “unscreened” were used interchangeably in the manuscript. I suggest using either the word "undiagnosed" or "unscreened" consistently across the manuscript. I prefer “undiagnosed”.

4. In the method section use the geographical areas from where the study population were selected clearly.

5. The authors used words like “incidence” (sentence 168); Risk Ratio (sentence 169); incidence rate ratio (sentence 174) which are wrong for this study as it is a cross-sectional study.

6. The primary dependent variable for this study is undiagnosed hypertension. However, the authors primarily focused on all hypertension (both diagnosed and undiagnosed) in all analyses. My observation and suggestion are as follows:

i. 38.5% is the overall prevalence of hypertension and half (50.9%) of them are unscreened. So, there are three categories of participants in respect to hypertension: 1) Normotensive, 2) undiagnosed hypertensive, and 3) diagnosed hypertensive. I suggest using a multinomial logistic regression with three category outcome to identify the associated factors of undiagnosed hypertension with normotensive as the reference category.

7. As the number of “no formal education” is low (Table 1), add them with primary education category.

8. In table 2, number of the normal BMI is missing.

9. In sentence 265, the authors mentioned the prevalence of undiagnosed hypertension is 50.9%. I guess this is the prevalence of undiagnosed hypertension among the hypertensive population. Be sure of it.

10. In the figure 1, the authors classified the hypertensive status as “unscreened”, “unaware”, “untreated”, “uncontrolled”, and “health risk”. However, these are not shown in the results, although these are important.

11. Figure 2 is not clear at all. The figure color and legend captions are not related. Either make it clear or delete it.

6. PLOS authors have the option to publish the peer review history of their article (what does this mean?). If published, this will include your full peer review and any attached files.

Reviewer #1: **Yes: **Dipak Kumar Mitra

---

## [Author Response · Author response to Decision Letter 0]

24 Aug 2023

Dear Dr. Bruno Pereira Nunes,

Thank you for considering our manuscript "BURDEN OF UNDIAGNOSTIC HYPERTENSION ANN ASSOCIATED FACTORS: A CHALLENGE FOR PRIMARY HEALTH CARE" for publication in PLOS ONE and for the feedback provided.

We greatly appreciate the time and effort the reviewer(s) and the academic editor dedicated to reviewing our manuscript. We have thoroughly addressed each of the comments and concerns raised and believe that the revisions have significantly improved the quality and clarity of the paper.

---

## [Decision Letter · Decision Letter 1]

19 Oct 2023

PONE-D-23-08509R1BURDEN OF UNDIAGNOSED HYPERTENSION AND ITS ASSOCIATED FACTORS: A CHALLENGE FOR PRIMARY HEALTH CARE IN URBAN COLOMBIAPLOS ONE

Dear Dr. Salazar Flórez,

Thank you for submitting your manuscript to PLOS ONE. After careful consideration, we feel that it has merit but does not fully meet PLOS ONE’s publication criteria as it currently stands. Therefore, we invite you to submit a revised version of the manuscript that addresses the points raised during the review process. The manuscript has been reassessed by two reviewers. The comments are appended below. We request authors to conduct a review of the comments providing detailed answers. 

We look forward to receiving your revised manuscript.

Kind regards,

Bruno Pereira Nunes, Ph.D.

Academic Editor

PLOS ONE

Reviewers' comments:

Reviewer's Responses to Questions

**Comments to the Author**

1. If the authors have adequately addressed your comments raised in a previous round of review and you feel that this manuscript is now acceptable for publication, you may indicate that here to bypass the “Comments to the Author” section, enter your conflict of interest statement in the “Confidential to Editor” section, and submit your "Accept" recommendation.

Reviewer #2: All comments have been addressed

Reviewer #3: (No Response)

2. Is the manuscript technically sound, and do the data support the conclusions?

Reviewer #2: Yes

Reviewer #3: (No Response)

3. Has the statistical analysis been performed appropriately and rigorously? 

Reviewer #2: Yes

Reviewer #3: (No Response)

4. Have the authors made all data underlying the findings in their manuscript fully available?

Reviewer #2: Yes

Reviewer #3: (No Response)

5. Is the manuscript presented in an intelligible fashion and written in standard English?

Reviewer #2: Yes

Reviewer #3: (No Response)

6. Review Comments to the Author

Reviewer #2: Please check the use of HTN (sometimes HNT is used in error).

Please clarify education level, it sometimes reads as the level of current education, but you maximum leel of education gained. Phrases such as 'people in primary school' can be misleading.

It would be nice to add a reference for the population projection (DANE).

Lines 316/317 - I am not sure you mean less health conscious, but maybe they feel they are healthier and probably are healthier so less likley to get checked or screened? I agree with the latter part that they are less susceptible to other diseases.

Line 371 - be more explicit. Familiy history was not accounted for as a covariate?

Reviewer #3: Thank you for the opportunity to review the manuscript “Burden of undiagnosed hypertension and its associated factors: a challenge for primary health care in urban Colombia”.

The introduction is to long, almost 3 pages. I suggest you be more succinct.

It doesn't make sense for the sentence "Health policymakers might require such empirical evidence to create targeted strategies for controlling HTN both in the country and the world" to be after the objectives. It fits in with the discussion rather than the introduction.

The sample size is not adequate given that at least 306 participants were supposed to be interviewed (not counting the 10% that is recommended to be added for losses and refusals), and only 286 were interviewed.

Please check all manuscript and avoid sentences like "hypertensive participants". Change to "participants with hypertension". The same for obese, avoid “were obese or overweight”. Instead, use had obesity of overweight.

In general, the manuscript provides interesting results that show a high rate of underdiagnosed cases of hypertension. The failure to reach 100% of the sample calculation needs to be discussed as a limitation. Pejorative terms such as hypertensive people and obese people need to be revised throughout the text. I would also suggest a grammatical review of the English, I believe that some sentences could be rewritten for better understanding.

7. PLOS authors have the option to publish the peer review history of their article (what does this mean?). If published, this will include your full peer review and any attached files.

Reviewer #2: **Yes: **Holly Pavey

Reviewer #3: No

---

## [Author Response · Author response to Decision Letter 1]

23 Oct 2023

Colombia, October 20th, 2023

Emily Chenette

Editor-in-Chief

PLOS ONE

Dear Editor Chenette and Reviewers,

We would like to extend our heartfelt gratitude for your constructive comments. They have played a pivotal role in refining our presentation of findings on this significant public health matter. Our responses to your observations are delineated below, and the corresponding changes within the manuscript are highlighted in yellow.

Reviewer 2: Comments Reply

Please check the use of HTN (sometimes HNT is used in error).

The acronym has been consistently standardized to "HTN" throughout the document. Please clarify education level, it sometimes reads as the level of current education, but you maximum level of education gained. Phrases such as 'people in primary school' can be misleading.

We have made it clear that we refer to the highest level of education attained.

It would be nice to add a reference for the population projection (DANE).

The DANE reference has now been incorporated. Lines 316/317 - I am not sure you mean less health conscious, but maybe they feel they are healthier and probably are healthier so less likley to get checked or screened? I agree with the latter part that they are less susceptible to other diseases.

We meant to indicate that individuals who exercise regularly often feel healthier and thus may be less concerned about regular check-ups. This has been clarified in the text.

Line 371 - be more explicit. Family history was not accounted for as a covariate?

We have made it explicit that family history was not included as a covariate.

Reviewer 3: Comments Reply

The introduction is to long, almost 3 pages. I suggest you be more succinct.

We've condensed the introduction for succinctness. It doesn't make sense for the sentence "Health policymakers might require such empirical evidence to create targeted strategies for controlling HTN both in the country and the world" to be after the objectives. It fits in with the discussion rather than the introduction.

The mentioned sentence has been relocated as suggested.

The sample size is not adequate given that at least 306 participants were supposed to be interviewed (not counting the 10% that is recommended to be added for losses and refusals), and only 286 were interviewed.

We have addressed the sample size limitation in the discussion. Please check all manuscript and avoid sentences like "hypertensive participants". Change to "participants with hypertension". The same for obese, avoid “were obese or overweight”. Instead, use had obesity of overweight.

Changes have been made to avoid pejorative terms, and the suggested terminologies are adopted.

In general, the manuscript provides interesting results that show a high rate of underdiagnosed cases of hypertension. The failure to reach 100% of the sample calculation needs to be discussed as a limitation. Pejorative terms such as hypertensive people and obese people need to be revised throughout the text. I would also suggest a grammatical review of the English, I believe that some sentences could be rewritten for better understanding.

We appreciate the feedback. The manuscript underwent thorough proofreading by a native English speaker, ensuring clarity and grammatical correctness.

Yours sincerely,

Jorge Emilio Salazar Florez

Research center coordinator

San Martin University Foundation, Sabaneta, Colombia

Ángela Patricia Echeverri

Student of medicine

San Martin University Foundation, Sabaneta, Colombia

Luz Stella Giraldo Cardona

Professor

San Martin University Foundation, Sabaneta, Colombia

---

## [Decision Letter · Decision Letter 2]

27 Oct 2023

BURDEN OF UNDIAGNOSED HYPERTENSION AND ITS ASSOCIATED FACTORS: A CHALLENGE FOR PRIMARY HEALTH CARE IN URBAN COLOMBIA

PONE-D-23-08509R2

Dear Dr. Salazar Flórez,

We’re pleased to inform you that your manuscript has been judged scientifically suitable for publication and will be formally accepted for publication once it meets all outstanding technical requirements.

Kind regards,

Bruno Pereira Nunes, Ph.D.

Academic Editor

PLOS ONE

Additional Editor Comments (optional):

Reviewers' comments:

Reviewer's Responses to Questions

**Comments to the Author**

1. If the authors have adequately addressed your comments raised in a previous round of review and you feel that this manuscript is now acceptable for publication, you may indicate that here to bypass the “Comments to the Author” section, enter your conflict of interest statement in the “Confidential to Editor” section, and submit your "Accept" recommendation.

Reviewer #2: All comments have been addressed

Reviewer #3: All comments have been addressed

2. Is the manuscript technically sound, and do the data support the conclusions?

Reviewer #2: Yes

Reviewer #3: Yes

3. Has the statistical analysis been performed appropriately and rigorously? 

Reviewer #2: Yes

Reviewer #3: Yes

4. Have the authors made all data underlying the findings in their manuscript fully available?

Reviewer #2: Yes

Reviewer #3: Yes

5. Is the manuscript presented in an intelligible fashion and written in standard English?

Reviewer #2: Yes

Reviewer #3: Yes

6. Review Comments to the Author

Reviewer #2: Thank you for addressing the comments, I am happy that the suggested comments have been addressed and changes mafe to the manuscript and therefore the manuscript has been much improved.

Reviewer #3: The authors made corrections and suggestions throughout the text. I now consider the manuscript approved for publication.

7. PLOS authors have the option to publish the peer review history of their article (what does this mean?). If published, this will include your full peer review and any attached files.

Reviewer #2: No

Reviewer #3: **Yes: **Felipe Delpino

---

## [Editor Report · Acceptance letter]

14 Nov 2023

PONE-D-23-08509R2 

BURDEN OF UNDIAGNOSED HYPERTENSION AND ITS ASSOCIATED FACTORS: A CHALLENGE FOR PRIMARY HEALTH CARE IN URBAN COLOMBIA 

Dear Dr. Salazar Flórez:

I'm pleased to inform you that your manuscript has been deemed suitable for publication in PLOS ONE. Congratulations! Your manuscript is now with our production department. 

Kind regards, 

on behalf of

Dr. Bruno Pereira Nunes 

Academic Editor

PLOS ONE